# Milk levels of transforming growth factor beta 1 identify mothers with low milk supply

**Rhea Sullivan**, **Alexandra Confair**, **Steven D. Hicks***

Department of Pediatrics, Penn State College of Medicine, Hershey, PA, United States of America

* Shicks1@pennstatehealth.psu.edu

## Abstract

Human milk is optimal for infant nutrition. However, many mothers cease breastfeeding because of low milk supply (LMS). It is difficult to identify mothers at risk for LMS because its biologic underpinnings are not fully understood. Previously, we demonstrated that milk micro-ribonucleic acids (miRNAs) may be related to LMS. Transforming growth factor beta (TGFβ) also plays an important role in mammary involution and may contribute to LMS. We performed a longitudinal cohort study of 139 breastfeeding mothers to test the hypothesis that milk levels of TGFβ would identify mothers with LMS. We explored whether TGFβ impacts the expression of LMS-related miRNAs in cultured human mammary epithelial cells (HMECs). LMS was defined by maternal report of inadequate milk production, and confirmed by age of formula introduction and infant weight trajectory. Levels of TGF-β1 and TGF-β2 were measured one month after delivery. There was a significant relationship between levels of TGF-β1 and LMS ($X^2 = 8.92$, p = 0.003) on logistic regression analysis, while controlling for lactation stage ($X^2 = 1.28$, p = 0.25), maternal pre-pregnancy body mass index ($X^2 = 0.038$, p = 0.84), and previous breastfeeding experience ($X^2 = 7.43$, p = 0.006). The model accounted for 16.8% of variance in the data (p = 0.005) and correctly predicted LMS for 84.6% of mothers (22/26; AUC = 0.72). Interactions between TGF-β1 and miR-22-3p displayed significant effect on LMS status (Z = 2.67, p = 0.008). Further, incubation of HMECs with TGF-β1 significantly reduced mammary cell number (t = -4.23, p = 0.003) and increased levels of miR-22-3p (t = 3.861, p = 0.008). Interactions between TGF-β1 and miR-22-3p may impact mammary function and milk levels of TGF-β1 could have clinical utility for identifying mothers with LMS. Such information could be used to provide early, targeted lactation support.

## Introduction

Human milk is considered the optimal form of nutrition for developing infants. Both the American Academy of Pediatrics and the World Health Organization recommend exclusive breastfeeding for the first six months and promote continued breastfeeding up to two years [1, 2]. Studies estimate that approximately 80 percent of infants in United States receive some human milk, but by 6 months, only half are breastfed, and only a quarter drink human milk

**Data Availability Statement:** FASTQ files from RNA sequencing involved in this project have been deposited into the NIH GEO database (GSE192543). GEO respository link: https://www.ncbi.nlm.nih.gov/geo/query/acc.cgi?acc=

GSE19254 (accessed on 25 December 2022). All additional data is available within S1 Table.

**Funding:** Gerber Foundation to SDH (#5295) and National Institutes of Health to RS (F30DA057094).

**Competing interests:** SDH serves as Chief Medical Officer for Quadrant Biosciences, and scientific advisory board member for Spectrum Solutions, neither of whom were involved in this research. AC has no conflicts of interest to report. RS has no conflicts of interest to report.

exclusively [3]. For most mothers, early breastfeeding cessation is unplanned [4]. A study of 1177 breastfeeding mothers found that 60 percent did not breastfeed their infant for as long as they desired, and concerns about lactation and milk supply were one of the primary causes of early breastfeeding cessation [5].

Despite the emphasis on exclusive breastfeeding, and the recognition that milk supply is a major driver of early breastfeeding cessation, there is a dearth of knowledge about the patho-physiology underlying human milk production [6]. Many studies examining lactation physiology involve animal models, with a strong focus on milk production as it relates to the dairy industry [7, 8]. Animal studies have established the importance of intracellular signaling for mammary proliferation and milk production [9, 10]. For example, studies in bovine epithelium show that transforming growth factor beta (TGF-β1) is crucial for apoptosis and involution of the mammary gland [11–13]. Animal studies also demonstrate that genetic variation can impact the volume and nutrient composition of milk [14], identifying specific genes such as *casein beta* (*CSN2*), *butyrophilin* (*BTN1A1*), and *milk fat globule epidermal growth factor-8* (*MFGE8*) [15, 16]. Epi-transcriptional factors, such as micro-ribonucleic acids (miRNAs) also display an association with mammary function in lactating women [17]. Our study of 160 lactating mothers found that levels of miR-22-3p and let-7g-5p in human milk were higher among those who experienced low milk supply (LMS) [18]. Intriguingly, TGF-β1 signaling is a putative target for both of these miRNAs.

The expression, function, and pathologic role of TGFβ has been extensively studied in lactation physiology [19, 20]. TGFβ regulates cell growth, apoptosis, cell differentiation, carcinogenesis and immune regulation within the mammary gland [21]. Expression of TGFβ isoforms varies widely across the stages of lactation, and is implicated in the pathology of mastitis and breast cancer [12, 22, 23]. Although the relationship between mammary function and TGFβ is well-described in animal models [20], to our knowledge, no studies have examined whether levels of TGFβ in human milk are associated with lactation efficiency or milk supply. Establishing a link between TGFβ levels and maternal risk for LMS could provide a novel way to identify women at risk for LMS and early breastfeeding cessation.

The primary goal of this study was to examine the relationship between levels of TGFβ in human milk and LMS. We hypothesized that elevated levels of TGFβ in the first month of lactation would be associated with risk of early, unplanned breastfeeding cessation due to LMS. This hypothesis was tested in a longitudinal cohort of 139 lactating mothers who planned to breastfeed beyond four months. A secondary goal was to explore whether interactions between TGFβ and LMS-related miRNAs (i.e., miR-22-3p and let-7g-5p) impact mammary involution. To achieve this secondary goal, we examined associations between TGFβ and miRNA levels in human milk, and interrogated the effects of TGFβ signaling on the proliferation and miRNA expression of cultured human mammary epithelial cells (HMECs).

## Materials and methods

### Study design

This prospective cohort study involved a convenience sample of 221 women, ages 19–42 years. The study was approved by the Institutional Review Board at the Penn State College of Medicine (STUDY00008657). All participants provided written informed consent. Enrollment occurred between April 2018 and October 2020. Eligible participants included mothers of full-term, singleton infants (37–42 weeks gestation), who planned to breastfeed beyond four months. Exclusion criteria included maternal morbidities that could impact ability to breastfeed (e.g. cancer, drug addiction, HIV), presence of neonatal conditions that could impact an infant's ability to feed (e.g. cleft lip/palate, metabolic disease, prolonged neonatal intensive care

admission >7 days), or factors that could impact longitudinal follow-up (e.g., plan to seek pediatric care at an outside medical center, plan for infant adoption). Participants were enrolled within seven days of delivery at the newborn nursery or the outpatient pediatrics clinic affiliated with our academic medical center. Longitudinal follow-up occurred at regularly scheduled infant check-ups (e.g., 1, 4, 6, and 12 months after delivery). Follow-up visits were successfully completed by 201/221 mothers (90.9%), and 139/221 (62.8%) contributed sufficient milk volume at the 1 month visit for TGFβ analysis.

## Primary medical outcome

The 139 participating mothers were dichotomized into two groups (LMS or non-LMS) based on the following criteria: As part of the infant feeding practices (IFP) survey (administered at 1, 4, 6, and 12 months), mothers were asked, "Have you had to supplement your child's diet with formula?" Participants who indicated formula had been introduced were asked, "Which of the following reasons best describes why you chose to introduce formula into your child's diet: a) Decreased or low breast milk production; b) My child showed signs of an allergy from breastfeeding; c) I chose to introduce formula for personal reasons (e.g. work, daycare, time restraints); d) Other". Mothers who indicated that formula was introduced due to decreased or low breast milk production were designated as LMS (n = 27). The 112 mothers who either maintained exclusive breast feeding throughout the duration of the study, or reported formula introduction for reasons other than "decreased or low breastmilk production" were included in the non-LMS group.

## Maternal characteristics

Medical and demographic characteristics for all participating mothers were collected through electronic surveys administered by research staff at enrollment. The following characteristics were assessed: maternal age (years), race (White, Black, Asian, Bi-Racial, Other), ethnicity (Hispanic or non-Hispanic), marital status, health insurance, maternal education, prior breastfeeding experience, pre-pregnancy body mass index (BMI; kg/m2), and maternal health status (presence/absence of chronic medical condition). Maternal nutrition was assessed using the Dietary Screener Questionnaire (DSQ) [24], a validated survey developed by the National Cancer Institute, which provides estimated daily intake of fruits (cups/day), vegetables (cups/day), dairy (cups/day), calcium (mg/day), and sugar (tsp/day).

## Infant characteristics

Infant delivery mode, gestational age at delivery, and sex were recorded. Infant feeding characteristics were collected through electronic administration of the modified IFP, completed by mothers 1, 4, 6, and 12 months after delivery. Results of longitudinal IFP survey responses were used to determine: 1) Duration of breastfeeding (defined as the period of time for which an infant received breastmilk on a daily basis); and 2) Infant age at the time of formula introduction. Infant weight (kg) and length (cm) were abstracted from the medical record at birth and one month post-delivery. For each infant, weight-for-length Z-score was determined at each time point using standardized curves from the World Health Organization, and the change in Z-score from birth to one month was calculated.

## Milk collection

One maternal breast milk sample, collected approximately one month after parturition, was used for this analysis. Milk (1–5 ml) was manually expressed from a sterilized nipple surface

into 50 ml RNAse-free tubes prior to feeding (i.e., fore-milk). To control for differences between breasts, mothers were instructed to express milk from the same breast at each time-point. At the time of milk collection, mothers reported presence/absence of nipple pain, and presence/absence of gastrointestinal or upper respiratory symptoms, as these factors could impact TGFβ levels. Time of day and time since delivery (days) was also recorded for each milk sample. Samples were immediately transferred to -20˚ C, underwent 1 freeze-thaw cycle for aliquoting, and were placed at -80˚ C while awaiting downstream analysis.

## Milk analysis

Milk samples (n = 139) were spun for 20 min at 4˚C at 200 rcf to separate lipid, cellular, and skim milk components. Skim milk was spun for an additional eight minutes at 16000g. As we have previously described [25], an automated immunoassay (ProteinSimple, San Jose, CA, USA) was used to measure levels of transforming growth factor beta 1 (TGF-β1) and transforming growth factor beta 2 (TGF-β2). After a standard sample activation procedure with HCl and NaOH/Hepes, a high-performance magnetic bead-based multiplex assay was employed per manufacturer instructions. Measurements were obtained on a Luminex MAG-PIX instrument with xPONENT software.

For samples with remaining lipid fraction (n = 137), miRNA levels were also measured, as we have previously reported [26]. Briefly, RNA was purified with a Norgen Circulating and Exosomal RNA Purification Kit (Norgen Biotech; Ontario, Canada) and sequenced at the SUNY Molecular Analysis Core using the Illumina TruSeq Small RNA Prep protocol on a NextSeq500 instrument (Illumina; San Diego, CA, United States) at 10 million, 50 base, paired-end reads per sample. Reads were aligned to the hg38 build of the human genome using Partek Flow (Partek; St. Louis, MO, United States) and the Bowtie2 aligner for quantification of mature miRNAs with miRBase v22. Quantile normalization was performed prior to statistical analysis.

## Human mammary epithelial cell (HMEC) culture

The impacts of TGFβ on miRNA expression and mammary growth were investigated using HMECs (Lonza Biosciences, Cat. # CC-2551). Mammary cells were plated at $8.75 \times 10^4$ cells/well on 12-well plates with tissue culture-treated plastic. Monolayer cells were cultured in Mammary Epithelial Growth Medium (Lonza Biosciences), supplemented with bovine pituitary extract, human epidermal growth factor, insulin, hydrocortisone, and gentamicin-amphotericin, according to manufacturer recommendations. TGF-β1 (R&D Systems, Cat. # 7754-BH-005/CF) was reconstituted in 0.1% bovine serum albumin in 4 mM hydrochloric acid. HMECs were cultured for 48 hours in the presence of 10 ng/ml TGF-β1 or vehicle. Daily media changes were not performed during incubation periods.

## Cell count assay

To assess the effect of TGF-β1 on HMEC growth, total cell numbers were counted in each well using the automated "Cell Count" brightfield feature on Cytation5 (Biotek) immediately prior to TGF-β1 exposure. After 48 hours of incubation with 10 ng/ml TGF-β1 or vehicle, HMEC counting was repeated. Change in cell count was calculated as the pre/post-treatment change HMEC count. Four biological replicates and three technical replicates were completed for each treatment group.

## HMEC miRNA expression

HMECs were harvested after 48 hours of incubation with 10 ng/ml TGF-β1 or vehicle using trypsin (Lonza Biosciences, Cat. # CC-5034). Timing of exposure and concentrations were chosen based off previous HMEC studies profiling gene expression changes after TGF-β1 incubations [27, 28]. Total RNA was extracted using the miRNeasy Micro Kit (Qiagen, Cat. # 217084). MicroRNAs were isolated, reverse transcribed, and amplified with the MystiCq microRNA PCR kit (Sigma, Cat. # MIRRM03). Custom miR-22-3p DNA primers were designed as previously described [29] (Integrated DNA Technologies, Coralville, Iowa). Let-7g-5p DNA MystiCq primers were purchased from Sigma (Cat. # MIRAP00011). Small nucleolar RNA C/D box 44 (SNORD44/U44) was used as an internal control gene [30]. qPCR was repeated in technical triplicate for all biological replicates.

## Statistical analysis

Medical and demographic characteristics were compared between LMS and non-LMS groups using a Student's t-test, Mann-Whitney U-test, chi-square test, or two-proportion z-test, as appropriate. Levels of TGF-β1 and TGF-β2 displayed non-parametric distribution and were compared between groups with a Mann-Whitney U-test. Secondary statistical analyses focused on TGF-β1, which displayed a difference between LMS and non-LMS groups. The relationship between TGF-β1 levels and maternal medical/demographic factors was assessed with a Spearman correlation test, or Mann Whitney U-test, as appropriate. A logistic regression analysis was used to assess the ability of milk TGF-β1 levels at 1 month to discriminate LMS and non-LMS groups, while controlling for confounding medical and demographic variables. Assumptions of non-collinearity were tested with variance inflation factors and the contributions of individual factors were determined with omnibus likelihood ratio tests. Overall model fit was assessed with Akaike Information Criterion (AIC). Discriminative ability of TGF-β1 was reported using area under the curve (AUC) on a receiver operating characteristic curve. Finally, we explored the relationship of TGF-β1 expression with other molecular markers of LMS. Interactions between LMS-related miRNAs (let-7g-5p, miR-22-3p) and TGF-β1 were assessed for their impacts on LMS status using a logistic regression with omnibus analysis of variance testing. Differences in miR-22-3p and let-7g-5p levels and changes in cell counts were compared between TGF-β1 and vehicle treatments using two-tailed student t-tests.

## Results

Participating mothers (n = 139) had an average age of 29.6 (± 4.8) years and an average pre-pregnancy BMI of 27.8 (± 7) kg/m$^2$ (Table 1). The majority were white (101/139, 72.7%), and non-Hispanic (123/139, 88.5%). Most were married (107/139, 77.0%), had a college diploma (88/139, 63.3%), and had private medical insurance (102/139, 73.4%). Nearly half reported at least one chronic medical condition (64/139, 46.0%), but few had depression (19/139, 13.7%). Most had previously breastfed (88/139, 63.3%). Most infants were delivered vaginally (109/139, 78.4%), and their average gestational age was 39 (± 1) weeks.

Nearly one in five mothers met criteria for LMS (27/139, 19.4%). Mothers with LMS were less likely to be feeding any breastmilk at 4 months (z = -1.79, p = 0.037), and those who were breastfeeding at four months were more likely to be supplementing with formula (z = 7.8, p < 0.001) (Fig 1A and 1B). Mothers with LMS were less likely to report they had breastfed as long as they wanted (z = -2.86, p = 0.004) (Fig 1C). They were also less likely to have breastfed previously (z = -2.27, p = 0.023) (Fig 1D). Infants of mothers with LMS displayed lower weight-for-length Z-scores at one month relative to infants of mothers without LMS (U = 1114, p = 0.020) (Fig 1E). There was no difference in maternal age, pre-pregnancy BMI, race,

**Table 1. Maternal, infant, and milk characteristics.**

| Characteristic | All (n = 139) | LMS (n = 27) | No LMS (n = 112) |
|---|---|---|---|
| *Maternal traits* | | | |
| Age (years), mean (SD) | 29.6 (4.8) | 30.0 (5.3) | 29.5 (4.7) |
| Race, n (%) | | | |
| Asian | 8 (5.8) | 2 (7.4) | 6 (5.4) |
| Bi-racial | 5 (3.6) | 2 (7.4) | 3 (2.7) |
| Black | 15 (10.8) | 6 (22.2) | 9 (8.0) |
| Other | 10 (7.2) | 1 (3.7) | 9 (8.0) |
| White | 101 (72.7) | 16 (59.3) | 85 (75.9) |
| Hispanic ethnicity, n (%) | 16 (11.5) | 5 (18.5) | 11 (9.8) |
| Married, n (%) | 107 (77.0) | 19 (70.3) | 88 (78.6) |
| Private health insurance, n (%) | 102 (73.4) | 20 (74.1) | 82 (73.2) |
| College diploma, n (%) | 88 (63.3) | 18 (66.7) | 70 (62.5) |
| Previously breastfed, n (%) | 88 (63.3) | 12 (44.4) | 76 (67.9)[a] |
| BMI (kg/m2), mean (SD) | 27.8 (7.0) | 28.7 (8.2) | 27.6 (6.7) |
| Maternal health condition, n (%) | 64 (46.0) | 14 (51.9) | 50 (44.6) |
| Maternal depression, n (%) | 19 (13.7) | 3 (11.1) | 16 (14.3) |
| Maternal diet | | | |
| Fruit (cups/day), mean (SD) | 1.19 (0.6) | 1.31 (0.7) | 1.16 (0.5) |
| Vegetables (cups/day), mean (SD) | 1.57 (0.4) | 1.64 (0.4) | 1.55 (0.3) |
| Dairy (cups/day), mean (SD) | 1.89 (0.7) | 1.92 (0.8) | 1.88 (0.6) |
| Calcium (mg/day), mean (SD) | 1049 (197) | 1040 (194) | 1051 (198) |
| Added sugar (tsp/day), mean, SD | 19.0 (8.3) | 21.2 (14.2) | 18.5 (6.1) |
| *Infant traits* | | | |
| Cesarean delivery, n (%) | 30 (21.6) | 4 (14.8) | 26 (23.2) |
| Gestational age (weeks), mean (SD) | 39 (1) | 39 (1) | 39 (1) |
| Male sex, n (%) | 61 (43.9) | 15 (55.6) | 46 (41.1) |
| Δ WfL Z-score (0–1 month), mean (SD) | 1.13 (1.5) | 0.6 (1.4) | 1.25 (1.5)[a] |
| *Milk-related factors* | | | |
| Breastfeeding duration (weeks), mean (SD) | 30 (14) | 22 (10) | 33 (14) |
| Earliest formula intro (weeks), mean (SD) | 28 (21) | 7 (6) | 33 (21) |
| Nipple pain, n (%) | 28 (20.7) | 4 (14.8) | 24 (22.2) |
| Maternal URI or GI Infection, n (%) | 5 (3.6) | 0 (0) | 5 (4.5) |
| Clock time of milk collection (24h), mean (SD) | 12 (4) | 13 (5) | 12 (3) |
| Infant age at milk collection (days), mean (SD) | 38 (13) | 39 (15) | 38 (13) |

[a]Denotes significant difference (p < 0.05) on t-test, U-test, or chi-square test. Abbreviations: Body mass index (BMI), Weight-for-length (WfL). Missing data for maternal age (n = 1, 0 LMS), maternal BMI (n = 5, 1 LMS), nipple pain (n = 4; 1 LMS), formula introduction (n = 12; 0 LMS), Time of milk collection (n = 28; 6 LMS).

ethnicity, education, marital status, health insurance, or chronic medical problems between groups. There was also no difference in delivery method or gestational age between groups.

Milk samples were collected, on average, 38 (± 13) days after delivery. The average time of collection was 12 pm (± 4 hours). Most mothers were pumping some milk at the time of collection (85/139, 66.4%). Few reported nipple pain (28/135, 20.7%), and few had an upper respiratory or gastrointestinal infection (5/139, 3.6%). There was no difference in date of collection (d = -0.07, p = 0.72) or time of sample collection (d = 0.19, p = 0.016) between groups. Those with LMS were not more likely to report nipple pain at the time of collection ($X^2$ = 0.56, p = 0.45), or report a current infection ($X^2$ = 1.25, p = 0.26).

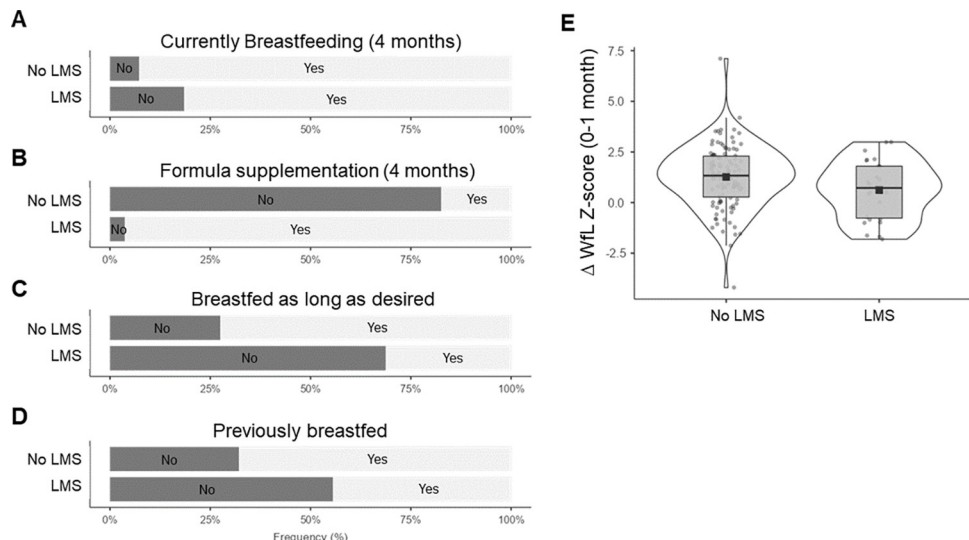

**Fig 1. Mothers with low milk supply display differential feeding patterns and infant growth.** The bar plots display breastfeeding patterns for mothers with (n = 27) and without (n = 112) low milk supply (LMS). Mothers with LMS were less likely to be breastfeeding four months after delivery (A; z = -1.79, p = 0.037), and they were more likely to have supplemented with formula (B; z = 7.8, p < 0.001) on two-proportion z-testing. Mothers with LMS were less likely to have breastfed as long as desired (C; z = -2.86, p = 0.004) and less likely to have previous breastfeeding experience (D; z = -2.27, p = 0.023). The violin plots (E) display the change in infant weight-for-length (WfL) Z-score between birth and one month. Infants of mothers with LMS displayed reduced gains in WfL Z-score (U = 1114, p = 0.020) compared with infants of mothers without LMS on Mann-Whitney U-testing. Mean (black squares), median (line), 95% confidence interval (boxes), and standard deviation bars are displayed.

Levels of TGF-β1 were higher in the milk of mothers with LMS (1055 pg ± 793) than in mothers without LMS (700 pg ± 584, U = 1070, p = 0.009) (Fig 2A). There was no difference in the levels of TGF-β2 between the LMS (9251 pg ± 11771) and non-LMS group (7269 pg ± 7145, U = 396, p = 0.32) (Fig 2B). Levels of TGF-β1 were associated with date of collection (R = -0.203, p = 0.016) and maternal pre-pregnancy BMI (R = 0.285, p = 0.001) (Fig 2C and 2D). TGF-β1 was not associated with time of milk collection, maternal age, or maternal consumption of fruits, vegetables, dairy, sugar, or calcium. TGF-β1 was not impacted by previous breastfeeding experience (U = 2080, p = 0.47), nipple pain at the time of milk collection (U = 1408, p = 0.62), pumping (1687, p = 0.48), or current maternal infection (U = 165, p = 0.055).

Logistic regression analysis confirmed a significant relationship between milk levels of TGF-β1 and LMS ($X^2$ = 8.92, p = 0.003), while controlling for date of collection ($X^2$ = 1.28, p = 0.25), maternal pre-pregnancy BMI ($X^2$ = 0.038, p = 0.84), and previous breastfeeding experience ($X^2$ = 7.43, p = 0.006). The model accounted for 16.8% of variance in the data (AIC = 127, p = 0.005). It correctly predicted LMS for 84.6% of mothers (22/26; AUC = 0.72, sensitivity = 0.846, specificity = 0.589).

Among participants for whom milk miRNA sequencing was available (n = 137), interactions between TGF-β1 and miR-22-3p (Z = 2.67, p = 0.008) and interactions between TGF-β1 and let-7g-5p (Z = -2.53, p = 0.011) both displayed significant effects on LMS status (Fig 3 and 3B). Higher levels miR-22-3p were associated with higher levels of TGF-β1 in women with LMS, whereas the opposite trend was observed for let-7g-5p. One miRNA not associated with putative TGF-β1 signaling [31], miR-151a-3p, was interrogated for effects on LMS status. Interactions between TGF-β1 and miR-151a-3p (z = 1.23, p = 0.220) did not show significant effects on LMS status (Fig 3C).

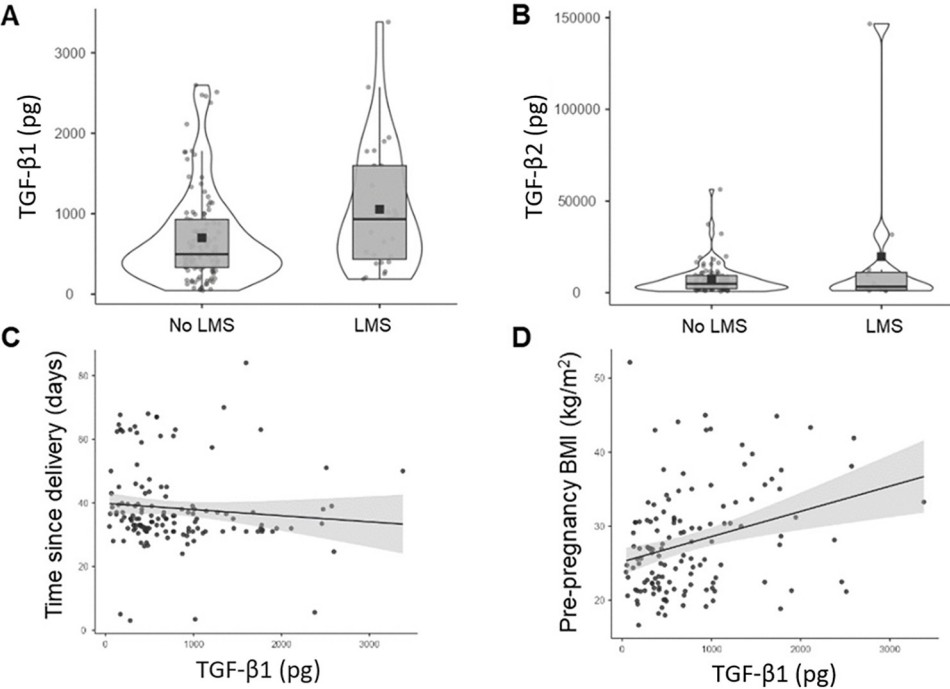

**Fig 2. Transforming growth factor beta-1 levels are higher in mothers with low milk supply.** The violin plots display milk levels of transforming growth factor beta (TGFβ)-1 (A) and TGF-β2 (B) in mothers with low milk supply (LMS; n = 27) and mothers without LMS (n = 112). Levels of TGF-β1 were higher in the milk of mothers with LMS (U = 1070, p = 0.009), but there was no difference in the levels of TGF-β2 between the LMS and non-LMS group (U = 396, p = 0.32) on Mann-Whitney testing. Levels of TGF-β1 displayed a significant relationship with lactation stage (measured as time since delivery in days) (C; R = -0.203, p = 0.016) and maternal pre-pregnancy BMI (D; R = 0.285, p = 0.001) on Spearman correlation testing. Trend lines with 95% confidence intervals are displayed.

Increases in mammary cell counts were suppressed by exposure to TGF-β1 (10 ng/ml) for 48 hours (t = -4.23, p = 0.003) (Fig 4A). HMECs exposed to TGF-β1 had significantly higher levels of miR-22-3p compared to vehicle controls (t = -3.86, p = 0.008) (Fig 4B). Let-7g-5p levels were unchanged between TGF-β1 and vehicle treatments (t = 0.6487, p = 0.5406) (Fig 4C).

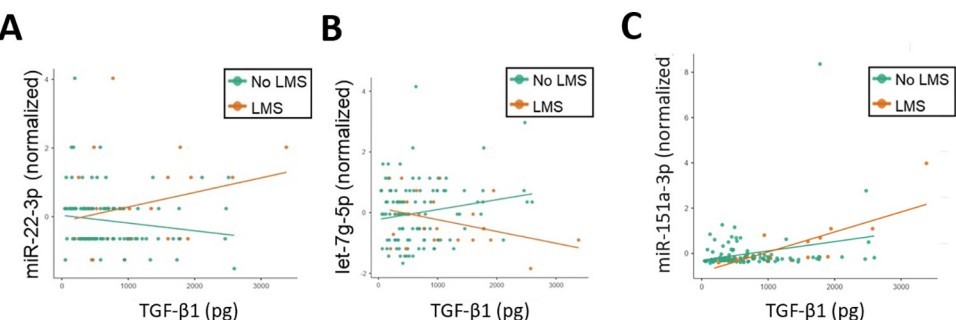

**Fig 3. Milk levels of transforming growth factor beta-1 are associated with microRNAs implicated in low milk supply.** The scatter plots display milk levels of TGF-β1 relative to quantile normalized levels of miR-22-3p (A) and let-7g-5p (B), two milk miRNAs previously associated with LMS. Interactions between TGF-β1 and miR-22-3p (Z = 2.67, p = 0.008) and interactions between TGF-β1 and let-7g-5p (Z = -2.53, p = 0.011) were both associated with LMS on logistic regression analysis. Interactions between TGF-β1 and miR-151a-3p, a miRNA that is not predicted to target TGF-β1 signaling, were not associated with LMS status (C). Trend lines for the LMS group are displayed in orange. Trend lines for the non-LMS group are displayed in green.

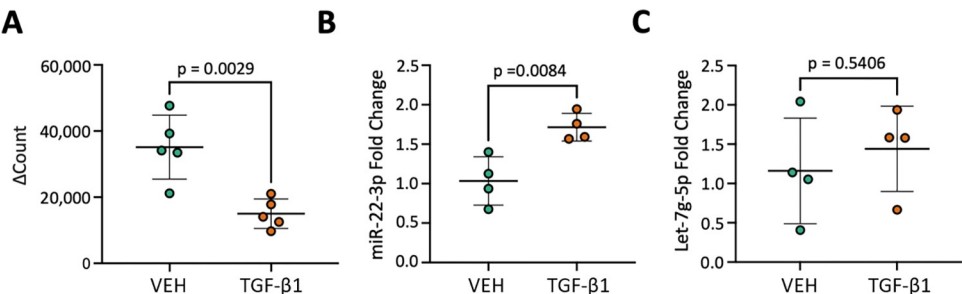

**Fig 4. TGF-β1 increases levels of miR-22-3p and decreases change in human mammary epithelial cell count.**
Human mammary epithelial cells (HMECs) were exposed to TGF-β1 (orange) or vehicle (green) for 48 hours. HMEC counts were suppressed in cultures treated with TGF-β1 (t = 4.22, p = 0.0029) (n = 5) (A). RT-qPCR results revealed a 1.71 fold change in miR-22-3p compared to vehicle treated controls (t = 3.86, p = 0.0084) (B). There was no change in let-7g-5p expression among HMECs exposed to TGF-β1 (t = 0.64, p = 0.54) (C). Statistical analysis was performed using a two-tailed Student's t-test.

## Discussion

Sustained, exclusive breastfeeding is a stated objective of the World Health Organization and the Centers for Disease Control and Prevention, but only 25% of U.S. infants are exclusively fed human milk at six months of age [3, 32]. LMS is a major reason why many mothers do not breastfeed as they planned [5]. Lactation support from trained professionals is an effective way to enhance breastfeeding rates [33], but resources are limited. The ability to objectively identify mothers at risk for LMS would allow for early, targeted lactation support, and potential improvement of breastfeeding rates. Although maternal factors, such as previous breastfeeding experience, provide some utility for predicting breastfeeding difficulty [34], our data suggest that biologic factors (i.e., TGF-β1) may enhance prediction of LMS. Such information may also be useful to mothers who experience guilt or frustration with lactation difficulties [35, 36]. In this cohort of 139 lactating mothers who intended to breastfeed beyond four months, milk levels of TGF-β1 (but not TGF-β2) were higher in those who experienced LMS. Few maternal factors appeared to impact TGF-β1 levels, aside from elevated maternal BMI and stage of lactation. After controlling for these factors, TGF-β1 continued to display a significant relationship with LMS, and its predictive utility for LMS exceeded that of previous breastfeeding experience.

These results also provide a conceptual framework for the molecular underpinnings of LMS. Previous studies have demonstrated the importance of TGF-β1 in mammary involution [11–13], and defined molecular targets through which TGF-β1 may influence lactation efficiency [19–21]. Here, we demonstrate in a large cohort of lactating women that TGF-β1 levels are directly associated with LMS, and identify specific miRNAs that may interact with TGF-β1 to impact milk supply. Interactions between TGF-β1 and LMS-associated miRNAs (miR-22-3p and let-7g-5p) displayed a significant association with LMS. Incubation of HMECs with TGF-β1 for 48 hours suppressed increases in the number of mammary cells and increased HMEC expression of miR-22-3p (but not let-7g-5p).

Our finding that TGF-β1 interacts with lactation-related miRNAs to influence LMS suggests that miRNAs may moderate the effects of TGF-β1 signaling on mammary involution. These findings are supported by the fact miR-22-3p has been experimentally confirmed to bind to the TGFβ type-1 receptor [37]. miR-22-3p has previously been shown to regulate apoptosis in other types of epithelium [38, 39]. Therefore, increases in miR-22-3p may represent a compensatory response to TGF-β1-induced apoptosis, particularly in the early phases of involution when infant feedings are still occurring. Studies in bovine models illustrate importance

of miRNAs in mammary involution [40], and research in lactating mothers has demonstrated that miRNAs are abundant in human milk and associated with milk production [18, 41–43].

There are several limitations of this study. We chose to define LMS based on maternal report rather than objective measurement of milk production or infant weights pre/post feed. Although these latter methods provide an accurate snapshot of lactation efficiency, they may undervalue maternal intuition, which may detect abrupt changes in milk production that can occur at any time during lactation. This notion is supported by our finding that infants of mothers who reported LMS displayed reduced weight-for-length Z-scores between birth and one month. Another limitation is that TGFβ was measured, on average, 38 days after delivery. To further assess the utility of TGFβ measurement for LMS detection, it would be useful to repeat TGFβ measurements longitudinally, beginning in the first week after delivery. We note that this cohort is predominantly White, non-Hispanic, and highly educated. Generalizability to other maternal populations should be confirmed. Finally, TGF-β1 levels measured within participant milk were around 1ng (~1000 pg), however, a concentration of 10 ng/ml was used for HMEC incubation experiments. This concentration of TGF-β1 was chosen due to the short incubation time frame, and previously published HMEC studies profiling gene expression changes [27, 28]. We chose to measure participant milk TGFβ levels using the Luminex MAG-PIX platform, which is a highly robust system to measure cytokines in multiplex. While this method is more expensive than traditional quantitative enzyme-linked immunosorbent assays (ELISAs), its bead-based system allows for greater sensitivity of cytokines with low concentrations than most commercially available ELISA kits and decreases labor and time spent using traditional methods. We recognize that this approach requires the access and funds needed for a multiplex immunoassay system.

In conclusion, this study demonstrates that levels of TGF-β1 in human milk are higher among mothers with LMS, and may influence mammary function through interactions with epi-transcriptomic mechanisms such as miR-22-3p. Further research is required to confirm that miR-22-3p binds TGF-β1-related transcripts to impact mammary involution, and determine whether TGF-β1 levels provide clinical utility for identifying mothers at risk for LMS.

## Supporting information

**S1 Table. Maternal, infant, and human milk data.**
(CSV)

## Acknowledgments

The authors thank Dan McKeone, MD (Penn State University) for assistance with cytokine analysis, Kaitlyn Warren, MS (Penn State University) for assistance with sample processing, Desirae Chandran, BS (Penn State University) for chart abstraction, and Claire Miller, BS (Penn State University) for data organization and project support.

## Author Contributions

**Conceptualization:** Steven D. Hicks.

**Data curation:** Alexandra Confair.

**Formal analysis:** Rhea Sullivan.

**Funding acquisition:** Rhea Sullivan, Steven D. Hicks.

**Investigation:** Rhea Sullivan, Steven D. Hicks.

**Methodology:** Steven D. Hicks.

**Project administration:** Alexandra Confair.

**Resources:** Alexandra Confair.

**Software:** Alexandra Confair.

**Supervision:** Steven D. Hicks.

**Writing – original draft:** Rhea Sullivan, Steven D. Hicks.

**Writing – review & editing:** Rhea Sullivan, Alexandra Confair, Steven D. Hicks.

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
