## [Decision Letter · Decision Letter 0]

13 May 2024

PONE-D-24-10053Milk levels of transforming growth factor beta-1 identify mothers with low milk supplyPLOS ONE

Dear Dr. Hicks,

Thank you for submitting your manuscript to PLOS ONE. After careful consideration, we feel that it has merit but does not fully meet PLOS ONE’s publication criteria as it currently stands. Therefore, we invite you to submit a revised version of the manuscript that addresses the points raised during the review process. The reviewers have provide their comments/suggestions for this manuscript:

- Reviewers pointed out several typographical errors and citation issues. For instance, Reviewer #1 highlighted errors in the naming of TGFβ1 and TGFβ2 on line 131, while Reviewer #2 noted the incorrect citation of reference 32. Correcting these errors will enhance accuracy.

- Reviewers requested clarification of specific methodology details. For example, Reviewer #2 mentioned that the collection of milk samples was inaccurately described as being done at each study visit (line 122). Clearly stating that samples were collected only once from each mother will improve methodological transparency.

- Reviewers suggested including additional information to strengthen clinical applicability. For instance, Reviewer #2 recommended providing a table of maternal characteristics to better correlate with clinical outcomes. This will make the study's findings more comprehensible and clinically meaningful.

We look forward to receiving your revised manuscript.

Kind regards,

Wan-Tien Chiang

Academic Editor

PLOS ONE

" ext-link-type="uri" xlink:type="simple">https://journals.plos.org/plosone/s/file?id=ba62/PLOSOne_formatting_sample_title_authors_affiliations.pdf"

2.Thank you for stating the following financial disclosure: 

"Gerber Foundation to SDH (#5295) and National Institutes of Health to RS (F30DA057094)" 

"SDH serves as Chief Medical Officer for Quadrant Biosciences, and scientific advisory board member for Spectrum Solutions, neither of whom were involved in this research. AC has no conflicts of interest to report. RS has no conflicts of interest to report. "

4. In the online submission form, you indicated that [FASTQ files from RNA sequencing involved in this project have been deposited into the NIH GEO database (GSE192543). GEO respository link: https://www.ncbi.nlm.nih.gov/geo/query/acc.cgi?acc=GSE192543 (accessed on 25 December 2022). Additional de-identified data may be made available with written request.]. 

Reviewers' comments:

Reviewer's Responses to Questions

**Comments to the Author**

1. Is the manuscript technically sound, and do the data support the conclusions?

Reviewer #1: Yes

Reviewer #2: Yes

Reviewer #3: Yes

2. Has the statistical analysis been performed appropriately and rigorously? 

Reviewer #1: Yes

Reviewer #2: Yes

Reviewer #3: Yes

3. Have the authors made all data underlying the findings in their manuscript fully available?

Reviewer #1: Yes

Reviewer #2: Yes

Reviewer #3: Yes

4. Is the manuscript presented in an intelligible fashion and written in standard English?

Reviewer #1: Yes

Reviewer #2: Yes

Reviewer #3: Yes

5. Review Comments to the Author

Reviewer #1: Line 43; please add reference.

Line 131; please fix the TGF□1 and TGF□2

Discussion: A gap remains in the discussion regarding the feasibility and cost-effectiveness of implementing TGF measurement. Addressing this gap would strengthen the overall impact and translatability of the research.

Reviewer #2: The study demonstrates great execution and writing, addressing a clinically significant topic supported by thorough bench research. The researchers have adhered to rigorous and comprehensive methodologies, with meticulous attention given to controls, including control vehicles and the assessment of miRNA expression differentials.

Certain enhancements could further improve the comprehensiveness of the study.

- Rectifying typing errors on line 131.

- Incorporating a table, if feasible, to outline maternal characteristics to enrich clinical correlation.

- Including a reference validating the Dietary Screener Questionnaire (DSQ).

- Clarifying in the methods section that milk samples were collected only once from each mother, rather than “at each study visit” as stated in line 122 to improve the accuracy of the study's methodology.

- Providing additional details regarding the number of milk samples analyzed (139 vs. 27) to derive the relationship patterns for TGF β1 measurements in Figure 2c and 2d to enhance transparency and interpretation of the findings. Was there any difference in patterns between mothers with and without LMS?

- Correction of reference 32 in line 278. This article does not attend to guilt and frustration in mothers with lactation difficulties. It is a strong point. May consider under mentioned references instead.

References:

- Jackson L, De Pascalis L, Harrold J, Fallon V. Guilt, shame, and postpartum infant feeding outcomes: A systematic review. Matern Child Nutr. 2021;17(3):e13141. doi:10.1111/mcn.13141

- Kam RL, Bennetts SK, Cullinane M, Amir LH. "I didn't want to let go of the dream": Exploring women's personal stories of how their low milk supply was discovered. Sex Reprod Healthc. Published online February 11, 2024. doi:10.1016/j.srhc.2024.100953

Reviewer #3: This is a well written, interesting piece of work. As a neonatologist working in a breastfeeding friendly NICU, I find it extremely important to begin to have an understanding of why some mothers struggle to produce enough breast milk to feed their babies. Further study may find that TGF B1 is a useful marker to identify these mothers early in the post-partum period.

6. PLOS authors have the option to publish the peer review history of their article (what does this mean?). If published, this will include your full peer review and any attached files.

Reviewer #1: No

Reviewer #2: No

Reviewer #3: No

---

## [Author Response · Author response to Decision Letter 0]

16 May 2024

We appreciate the reviewers’ time and expertise. We have incorporated all of their suggested minor revisions into the updated manuscript. We have added an additional table with participant characteristics and fixed several minor spelling errors. We have updated the funding statement to note that the funders had no role in study design, data collection and analysis, decision to publish, or preparation of the manuscript. We have updated our competing interest statement to note, “This does not alter our adherence to PLOS ONE policies on sharing data and materials”. We have made all of our data publicly available in a new supplemental table. A point-by-point response to reviewers is below:

Reviewer #1: 

Line 43; please add reference.

We have added two references supporting this statement. 

Line 131; please fix the TGF□1 and TGF□2

We have corrected the TGF beta nomenclature throughout the text and figures.

Discussion: A gap remains in the discussion regarding the feasibility and cost-effectiveness of implementing TGF measurement. Addressing this gap would strengthen the overall impact and translatability of the research.

Thank you for this excellent suggestion. We’ve added text to the discussion about the feasibility and cost-effectiveness of TGF measurement. This should add transparency about the strengths and limitations of this approach.

Reviewer #2: 

The study demonstrates great execution and writing, addressing a clinically significant topic supported by thorough bench research. The researchers have adhered to rigorous and comprehensive methodologies, with meticulous attention given to controls, including control vehicles and the assessment of miRNA expression differentials. Certain enhancements could further improve the comprehensiveness of the study:

- Rectifying typing errors on line 131.

These errors have been corrected. 

- Incorporating a table, if feasible, to outline maternal characteristics to enrich clinical correlation.

We have added Table 1, describing maternal, infant, and milk factors. 

- Including a reference validating the Dietary Screener Questionnaire (DSQ).

This reference has been added.

- Clarifying in the methods section that milk samples were collected only once from each mother, rather than “at each study visit” as stated in line 122 to improve the accuracy of the study's methodology.

We apologize for this oversight. The revised manuscript clarifies that only one sample was analyzed for each mother. 

- Providing additional details regarding the number of milk samples analyzed (139 vs. 27) to derive the relationship patterns for TGF β1 measurements in Figure 2c and 2d to enhance transparency and interpretation of the findings. Was there any difference in patterns between mothers with and without LMS?

Thank you for this suggestion. The legend for Figure 2 now includes the number of samples used to derive the relationships for TGFβ1. All of the mothers in the LMS group (27/27) and non-LMS group (112/112) provided a sample for analysis. The levels of TGFβ1 were higher in mothers with LMS (U = 1070, p = 0.009).

- Correction of reference 32 in line 278. This article does not attend to guilt and frustration in mothers with lactation difficulties. It is a strong point. May consider under mentioned references instead.

Jackson L, De Pascalis L, Harrold J, Fallon V. Guilt, shame, and postpartum infant feeding outcomes: A systematic review. Matern Child Nutr. 2021;17(3):e13141. doi:10.1111/mcn.13141

Kam RL, Bennetts SK, Cullinane M, Amir LH. "I didn't want to let go of the dream": Exploring women's personal stories of how their low milk supply was discovered. Sex Reprod Healthc. Published online February 11, 2024. doi:10.1016/j.srhc.2024.100953

We apologize that our reference list was misaligned with the text in our original submission. We have corrected this error and added the two references above. 

Reviewer #3: 

This is a well written, interesting piece of work. As a neonatologist working in a breastfeeding friendly NICU, I find it extremely important to begin to have an understanding of why some mothers struggle to produce enough breast milk to feed their babies. Further study may find that TGF B1 is a useful marker to identify these mothers early in the post-partum period.

Thank you. We agree that this is an important field of study!

---

## [Editor Report · Decision Letter 1]

30 May 2024

Milk levels of transforming growth factor beta 1 identify mothers with low milk supply

PONE-D-24-10053R1

Dear Dr. Hicks,

We’re pleased to inform you that your manuscript has been judged scientifically suitable for publication and will be formally accepted for publication once it meets all outstanding technical requirements.

Kind regards,

Wan-Tien Chiang

Academic Editor

PLOS ONE
---

## [Editor Report · Acceptance letter]

3 Jun 2024

PONE-D-24-10053R1 

PLOS ONE

Dear Dr. Hicks, 

I'm pleased to inform you that your manuscript has been deemed suitable for publication in PLOS ONE. Congratulations! Your manuscript is now being handed over to our production team.

Kind regards, 

on behalf of

Dr. Wan-Tien Chiang 

Academic Editor

PLOS ONE